# Construction and Swelling Properties of Thermosensitive N-isopropyl Acrylamide Microspheres With Controllable Size

**DOI:** 10.3390/ma12152428

**Published:** 2019-07-30

**Authors:** Chen Wang, Si-xian Lu, Liang Wang, Yao Hui, Yan-ru Lu, Wei-jia Chen

**Affiliations:** 1School of Materials Science and Engineering, Xi’an Polytechnic University, Xi’an 710048, China; 2School of Materials Science and Engineering, Xi’an Jiaotong University, Xi’an 710049, China

**Keywords:** N-isopropylacrylamide, thermosensitive microspheres, microspheres size, swelling behavior

## Abstract

In recent years, thermosensitive poly(N-isopropylacrylamide) (PNIPAM) microspheres have received extensive attention due to their many advantages, and their size and swelling ratio are two crucial factors. In this paper, homogeneous and hollow thermosensitive microspheres were prepared by free radical polymerization in an aqueous solution. The effects of the process parameters on the size of the microspheres were studied. The results indicated that the change in size during reaction at different temperatures was not obvious. The size of the microspheres ranged from 802 ± 35.4 nm to 423 ± 33.7 nm with the changes in the dosage of the initiator. Meanwhile, it was observed that the size of microspheres was slightly reduced due to the increase of reaction time. When the dosage of methyl methacrylate (MAA) is increased, the size of the hollow microspheres increased by more than 110%. The average size of the microspheres was smaller when the content of sodium dodecyl sulfate (SDS) was 3 wt%. The microspheres with varying reaction parameters showed a continuous decreasing swelling ratio when the temperatures were changed from 28 °C to 35 °C. In comparison with homogeneous microspheres, the average swelling ratio of hollow microspheres was larger.

## 1. Introduction

Thermosensitive poly(N-isopropylacrylamide) (PNIPAM) hydrogel microspheres have intriguing potential applications in switching intelligent systems [1,2], optical, photonic materials [3,4], and biological materials [5,6]. Known to many, PNIPAM forms swollen hydrogels of cross-linked species due to the existence of both hydrophilic amide groups and hydrophobic isopropyl groups in its side chains [7]. The PNIPAM-based hydrogels showed a negative thermal response around 32 °C, which is called the lower critical solution temperature in a linear polymer [8].

The reported preparation methods of PNIPAM hollow microspheres included the template method [9], inverse emulsion polymerization [10], in situ interfacial polymerization [11], Shirasu Porous Glass (SPG) membrane emulsification [12], and one-pot method [13]. The shrinking and swelling process of the hydrogel microspheres is controlled by diffusion transport, and thus the response rate is inversely proportional to the square of the smallest dimension of the gel [14]. Hence, PNIPAM microspheres have attracted a great deal of interest because of their small size and fast stimuli-responsive behavior, proving that it has got great application potentials [15,16]. The traditional methods used to prepare emulsions often include mechanical stirring, homogenizing, and spraying techniques. It is difficult to control the size of the obtained microspheres. These methods are not simple enough to control the size of the obtained microspheres. For thermosensitive PNIPAM microspheres used in drug delivery system, size controllability and narrow size distribution are two important properties. The size of the microspheres has a great effect on their distribution in the body and could also affect the interaction between the particles and cells [17]. The swelling ratio of the microspheres is one of its essential features that can be used to characterize the properties of a polymer network. The hydrogel microspheres can change their swelling behaviors and size in response to reaction conditions [18,19].

Based on these, the preparation of hydrogel microspheres with a controllable particle size is of great theoretical and practical value. In this paper, we attempted the design and synthesis of two types of thermosensitive microspheres (homogeneous and hollow) that were prepared by free radical polymerization in an aqueous solution from N-isopropylacrylamide (NIPAM) and methyl methacrylate (MAA) monomers. At the same time, we studied the effects of reaction temperature, initiator dosage, reaction time, and stirring speed on the size of the homogeneous microspheres. The effects of the dosage of MAA and SDS on the size of hollow microspheres were also discussed. For comparison, microspheres with different reaction parameters were prepared for the temperature-sensitive swelling tests. Compared with homogeneous microspheres, the average swelling ratio of hollow microspheres was improved. Presented in this work is as an example of the development of drug delivery systems. The current research results show that the two types of thermo-sensitive microspheres have great application prospects as controlled release drug delivery.

## 2. Materials and Methods

### 2.1. Chemicals

Monomers N-isopropylacrylamide (NIPAM), Methyl methacrylate (MAA), the crosslinker N,N-methylenebisacrylamide, surfactants sodium dodecyl sulfate (SDS), and initiator potassium persulfate were purchased from J & K Scientific Ltd., Beijing, China. Ammonia water (NH_3_∙H_2_O) was purchased from Sinopharm Chemical Reagent Co., Ltd., Shanghai, China. All other chemicals and solvents used in this paper were of analytical grade and solutions were prepared from distilled water.

### 2.2. Synthesis of Microspheres

Homogeneous microspheres were synthesized by the reaction process, which was described previously, with some modifications [20]. NIPAM (8.84 mmol), deionized water, and N,N-methylenebisacrylamide (0.32 mmol) were reacted in nitrogen atmosphere. This reaction occurred in a heated vessel under mechanical stirring and the temperature was increased slowly. After a period of time, potassium persulfate (0.11 mmol) was added as an initiator. Then, it underwent a continuous heating for 6 h at a temperature of 70 °C. Finally, the obtained mixture was washed three times with deionized water and centrifuged at a speed of 5000 r∙min^−1^ in the same centrifugal filter. After centrifugation of the solution, homogeneous microspheres were obtained. 

The hollow microspheres were obtained as follows: NIPAM (9.46 mmol), deionized water, and MAA (1.39 mmol) were reacted in nitrogen atmosphere. The reaction gradually increased from room temperature to 70 °C. Then potassium persulfate (0.11 mmol) was added. When the system solution turned to light blue, SDS (0.16 mmol) and N,N-methylenebisacrylamide (0.32 mmol) were added to the above reaction, and then the resulting solution was continuously heated for 10 h at 70 °C. Then the reaction was stopped and the solution was cooled down to room temperature. Afterwards, the pH of the solution was adjusted slowly to 9.0 by adding dropwise NH_3_∙H_2_O (1.0 M). The solutions were purified by dialysis for 72 h. The structure of microspheres is shown in Figure 1.

### 2.3. Measurements

Fourier transform infrared spectroscopy (FT-IR) spectra of the synthesized polymers were measured on a TENSOR FT-IR spectrometer (Bruker, Ettlingen, Germany) in the 4000–400 cm^−1^ region. Before testing, the microspheres were treated by lyophilization. The KBr pellets of samples were prepared by mixing 1 mg of samples and finely grounding with 100 mg KBr. The microspheres were washed repeatedly and diluted with anhydrous ethanol. The solutions were subjected to 30 min of ultrasonic oscillation. The suspension of the microspheres prepared in Section 2.2 was dropped on a copper mesh grid with holey carbon films and then dried at room temperature. The average size of the microspheres in the dry state was then analyzed by Transmission Electron Microscopy (TEM, JEN-200CX, Tokyo, Japan) and Nano Measurer software (Fudan University, Shanghai, China).

## 3. Results and Discussion

### 3.1. FT-IR and TEM

It can be seen from Figure 1c that the broad absorption peaks centered at 3309 cm^−1^ and 3304 cm^−1^ were attributed to N–H absorption peaks. The absorption peaks at 2977 cm^−1^ and 2971 cm^−1^ correspond to the methyl C–H stretching vibration bands. In particular, the strong sharp bands at 1654 cm^−1^ and 1649 cm^−1^ correspond to carbonyl (C=O) absorption peaks. The peaks at 1393 cm^−1^ and 1379 cm^−1^ could be ascribed to characteristic vibrations of –C(CH_3_)_2_. To our expectation, the peak at 1718 cm^−1^, which appeared only in the spectrum of hollow microspheres, was due to the existence of C=O in PMAA. The remaining MAA in the reaction system has been removed by dialysis. In general, there were small fluctuations in the characteristic absorption peaks. The microspheres structure is consistent with the results obtained from FT-IR analysis [21]. The transmission electron microscopic (TEM) images of microspheres are shown in Figure 1d,e.

### 3.2. Effects of Main Reaction Parameters on the Size of Homogeneous Microspheres

#### 3.2.1. Reaction Temperature

As shown in Figure 2a, the change in the size of microspheres at different reaction temperatures was not obvious. The size of microspheres ranged from 423 ± 33.7 nm to 872 ± 59.4 nm when the temperature increased from 60 °C to 90 °C. Since the half-life of KPS decreases with increasing temperature, it is 7.7 h at 70 °C, and it takes a longer time below 60 °C [22]. The microspheres were not obtained after 24 h of reaction at lower than 60 °C. A higher temperature produced smaller microsphere sizes within a certain temperature range. Increasing the temperature apparently led to an increase in the decomposition rate of the initiator, which resulted in an enhancement of the number of polymerization loci, and the particles could not grow normally, resulting in uneven particle size [23]. In other words, a typical polymerization temperature was 70 °C.

#### 3.2.2. Initiator Dosage

Homogeneous microspheres were prepared with different initiator dosages. The size of the microspheres is shown in Figure 2b. With the increasing of the dosage of the initiator, the amount of free radicals in the reaction system increased. Obviously, the size of microspheres ranged from 802 ± 35.4 nm to 423 ± 33.7 nm when the initiator dosage was increased from 1 wt% to 3 wt%. On the contrary, with the increasing initiator dosage (3 wt%–7 wt%), the microspheres diameter increased, while the microsphere morphology became smaller. In the early stage of the reaction, the active chain formed by the initiation was also increased [24]. Using this method, it was easy to generate larger-sized microspheres (1308 ± 99.6 nm). Based on these results, we could obtain a smaller size microsphere when the initiator dosage was 3 wt%.

#### 3.2.3. Reaction Time

As shown in Figure 2c, the size of microspheres ranged from 608 ± 30.1 nm to 423 ± 33.7 nm when the reaction time was increased from 4 h to 9 h. It was observed that the size of the microspheres decreased slightly as the reaction time increased because of the small size and poor stability of particles formed in the initial stage of the reaction. These particles had a strong tendency to agglomerate, which led to larger particles. As the reaction proceeds, the particles grew up gradually. Then the attached microspheres were easily detached from the agglomerated particles, which improved the uniformity of microspheres.

#### 3.2.4. Stirring Speed

As shown in Figure 2d, the size of the homogeneous microspheres ranged from 453 ± 85.1 nm to 530 ± 19.3 nm. No significant effect of the stirring rate during preparation on microspheres was observed. However, it had a great effect on the size uniformity of microspheres. As the stirring speed increased, the probability of collision between microspheres increased. It was easier to produce smaller-size microspheres, but their uniformity was poor [25]. On the whole, we could obtain smaller size microspheres with a stirring speed of 600 r∙min^−1^.

### 3.3. Effects of Main Reaction Parameters on the Size of Hollow Microspheres

As shown in Figure 3, the effects of the dosage of MAA and SDS on the size of the hollow microspheres were studied. In the absence of the SDS, the size of the microspheres was larger than the other conditions studied here. Instead, in the average dosage of MAA, the size of the microspheres increased by more than 110%. Nevertheless, the average size of microspheres decreased over 70% as the concentration of SDS increased. During the present study, the charge on the surface of the core particles that were formed by the copolymers of NIPAM and MAA was mainly derived from MAA. It was assumed that an increase in the MAA dosage might be caused by a decrease in pH of solution, thereby reducing the charge density of the surface of the core particles. Unstable particles tended to aggregate spontaneously, leading to nuclear enlargement. SDS could greatly reduce the average aggregation number of PNIPAM oligomers. Owing to the obvious synergistic stabilization of SDS, nuclear particles were easily stabilized in the system. Thus, microspheres became smaller as the size of the core decreased.

### 3.4. Swelling Behavior

The gravimetric method was used to study the microsphere swelling ratio [26]. As expected, microspheres with varying reaction parameters showed a continuous decreasing in the swelling ratio when the temperatures was changed from 28 °C to 35 °C, induced by the thermally sensitive phase transition of the microspheres (Figure 4). It could also be concluded that the major collapse of microspheres occurred in the temperature range of 31–32 °C.

The overall trend indicated that the homogeneous microspheres with a higher reaction temperature had a higher swelling ratio. When the reaction temperature was maintained at 85 °C, the swelling ratio of microspheres has the maximum value of 22.0. Besides that, the swelling ratio of microspheres that were prepared at different initiator dosage was measured. The results show that homogeneous microspheres had the highest swelling ratio when 3 wt% of the initiator was utilized. Moreover, it was clear that the swelling ratio of the microspheres with different reaction times from large to small were 15.4, 14.4, 13.0, 12.0, 8.5, and 1.0. Especially, the swelling ratio of the microspheres with the reaction time of 6 h decreased gradually from 14.4 to 0.25. For constant stirring, with the increase of the stirring speed, the swelling ratio of the microspheres changed from 16.4–0.1. However, the swelling ratio of microspheres was lower when the stirring speed was up to 1000 r∙min^−1^.

As can be seen from Figure 5, the change of the swelling ratio of the hollow microspheres was not obvious at 28 °C. The swelling ratio fluctuated around 20. Compared with homogeneous microspheres, the average swelling ratio of the hollow microspheres was improved.

## 4. Conclusions

In conclusion, two types of thermosensitive microspheres had been successfully prepared from readily available starting monomers with a convenient procedure. It was found that the size and swelling ratio of the microspheres were controlled with different reaction parameters. Compared with homogeneous microspheres, the average swelling ratio of the hollow microspheres was improved. Homogeneous and hollow thermosensitive microspheres have great application prospects as a controlled release drug delivery vehicle.

## Figures and Tables

**Figure 1 materials-12-02428-f001:**
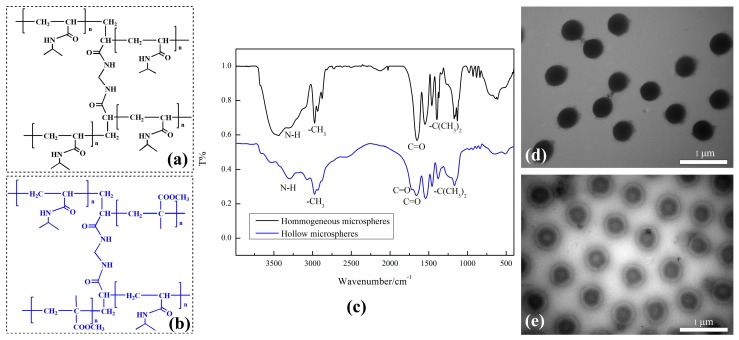
Characterizations: (**a**) structure of homogeneous microspheres; (**b**) structure of hollow microspheres; (**c**) Fourier transform infrared spectroscopy (FT-IR) spectra; (**d**) Transmission Electron Microscopy (TEM) images of homogeneous microspheres; (**e**) TEM images of hollow microspheres.

**Figure 2 materials-12-02428-f002:**
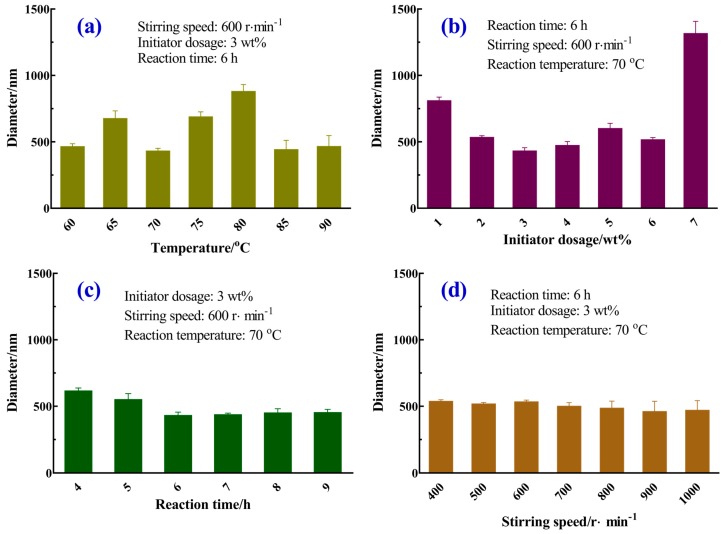
Effect of reaction parameters on the diameter of homogeneous microspheres: (**a**) reaction temperatures, (**b**) initiator dosages, (**c**) reaction time, and (**d**) stirring speed.

**Figure 3 materials-12-02428-f003:**
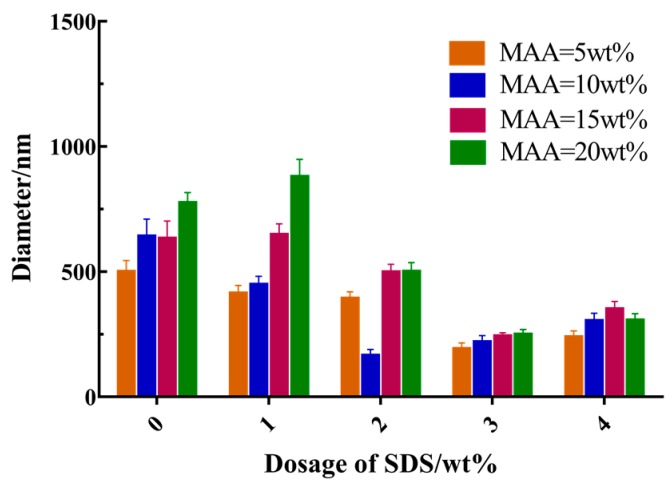
Effect of reaction parameters on the diameter of the hollow microspheres.

**Figure 4 materials-12-02428-f004:**
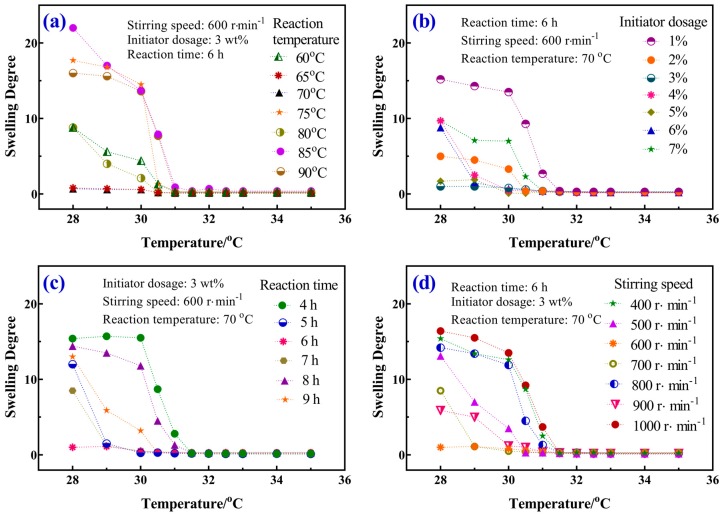
Effect of reaction parameters on the swelling ratio of homogeneous microspheres: (**a**) reaction temperatures, (**b**) initiator dosages, (**c**) reaction time, and (**d**) stirring speed.

**Figure 5 materials-12-02428-f005:**
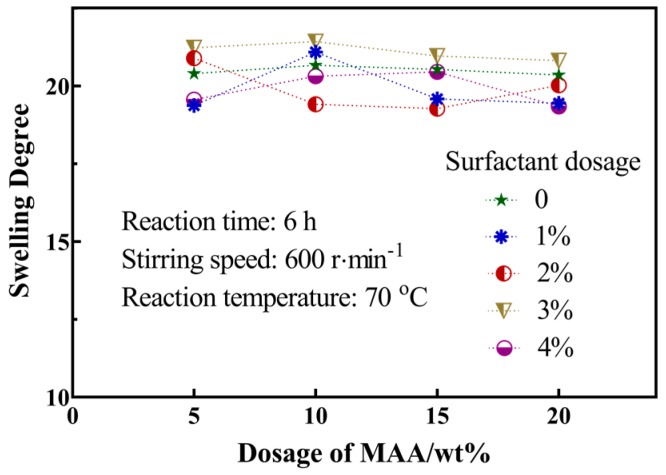
Effect of reaction parameters on the swelling ratio of the hollow microspheres.

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
