# Peer review of "Construction and Swelling Properties of Thermosensitive N-isopropyl Acrylamide Microspheres With Controllable Size"

_materials, 2019, doi:10.3390/ma12152428_

Reviewer 1 Report

Now, this paper may be published

Author Response

Dear Editor and Reviewers: Thank you for your letter and for the reviewers’ expert comments on our manuscript entitled “Construction and Swelling Properties of Thermosensitive N-isopropyl Acrylamide Microspheres with Controllable Size” (ID: materials-556092). All those comments are quite valuable and very helpful for further revising and improving the present manuscript and also guiding our future researching focuses. According to these expert comments, we have made the corresponding corrections carefully as follows. Certainly, these corrections do not influence the content and researching intention of the manuscript. Thank you for your attention and kind help. Best wishes Liang Wang Response to Reviewer 1 Comments Point 1: English language and style are fine/minor spell check required. Now, this paper may be published. Response 1: According to the Reviewer’s comments, we have made the corresponding corrections carefully in the manuscript. Certainly, these corrections do not influence the content and researching intention of the manuscript.

Reviewer 2 Report

This communication entitled “Construction and Swelling Properties of Thermosensitive N-Isopropylacrylamide Microspheres with Controllable Size” by Wei-Jia Chen et al. concerns the synthesis of thermo-responsive pNIPAM microgels as well as hollow pNIPAM microgels. After synthesis and characterization (performed by TEM and FT-IR analysis) authors modified a series of parameters in order to control particle dimension and swelling properties for pure and swelling degree for hollow microgels. Parameters as polymerization temperature, reaction time, stirring speed or amount of initiator were modified in order to study its influence in particle size and swelling rate.

After careful evaluation I consider that this work in not appropriate to be published in Materials. The principal reason to be rejected is that authors need to prove, demonstrate and support the results they obtained by, for example, including previously reported references. They should provide more probes in order to support their conclusions, overall after the enormous number of results they provide, taking into account that is a communication. Basically, in this work authors describe their results and take their own conclusions, but without supporting, it is essential that they compared with other ones.

Initially, in the abstract, authors says: “microspheres have received extensive attention due to their many advantages, but their size and swelling ratio are two uncontrollable factors”. I am pretty sure that these parameterers can be easily controlled. I do not know whay author can use the word “uncontrollable”. I have expertise with pNIPAM microgels and I can say that the size and swelling properties can be totally controlled, and a lot of literature is extensively reported. Indeed, there is some parts that are not logical, as example, what means: “The results indicated that the change in size during reaction at different temperatures was not obvious”, why is not obvious?. At the end of the Abstract authors says that the average swelling ration of hollow microspheres were improved, but how??, I mean, in which sense they were improved??.

Can authors say what means SPG in line 37?

In line 43 authors says: “it is difficult to control the size of the obtained microspheres”. It is not difficult to control the microgel size, for example by adjusting parameters as monomer amount or surfactant concentration or even the cross-linking density has been reported to control the particle size.

In Materials and Methods section authors must include in which transmission electron microscope they obtained the TEM images, and how they prepared the TEM grids, this is important information. Indeed, the FTIR also disappeared in terms of equipment used for analysis and sample preparation.

In the section 2.2 authors should briefly describe how they prepared the microgels, instead of only mention reference 20. For hollow particles, what is the role of the MAA??, how the hollow structure is formed, after increase the pH??. What is the dimension of the hollow. Can authors control this parameter, the size of the hollow produced into the microgel??

Concerning the FT-IR, if the MMA is eliminated in order to provide hollow structure, why is still in the microgel, as is confirmed by the C=O band at 1718 cm-1? In line 94 authors said: “The microsphere structure is consistent with the results obtained from FTIR analysis”. There is a lot of literature concerning FTIR than can support this comment, however no references are included.

Concerning the influence in the reaction temperature, authors says: “Since the half-life of KPS decreases with increasing temperature, it is 7.7 h at 70ºC”, authors must included some reference to prove it, and also “and takes longer time below 60ºC”, I would like to know what longer time means, it is necessary be more specific, and try to prove the reported data. Why the particles size decreases when the polymerization temperature increases above 70ºC, why decreases at 85ºC if is compared with that at 80ºC?? Why the study was performed at 600 rpm, 3% initiator and 6h?, There is no explanation in the manuscript

For the study at different initiator quantities, in line 112 authors says “Obviously, the size of microspheres ranged from 802 ± 35.4 nm to 423 ± 33.7 nm when the initiator dosage was increased from 1wt% to 3wt%. What is so obvious??. Indeed, the reasons about why the tendency for increasing the size with the initiator dosage 3 wt%-7wt% should be supported by including literature.

Regarding the reaction time, why author chose a range between 4 and 9 h? Indeed, the reasons that author provide about the tendency to agglomerate, leading to large particles should be demonstrated for example by measuring the surface particle charge.

About stirring investigations, authors conclude that as the stirring speed increased, “the probability of collision between microspheres increased. It was easier to produce smaller-size microsphers”. The author must to include any reported reference to confirm this affirmation?

Concerning the effects of main reaction parameters on the size of hollow microspheres, I would like to know if the amount of MAA influences the hollow size. Apart from that, it is well-known that the SDS amount reduce the particle size during pNIPAM polymerization, consequently authors do not provided any novelty with this investigation. Indeed, the dependence of MAA in particle size is different at 2 wt% compared with the other percentages, what is the reason for this different behavior?

And finally, concerning the swelling behavior there is a plenty of results with no explanation, basically authors describe the results with almost no explanation or if there so kind of explanation it is not supported by literature or previous publications.

In summary, I think that this paper to be accepted, authors should organize the results in a better way and try to find in literature their obtained results, supporting their results and conclusion with previously reported literature, that in case of pNIPAM is enormous

Author Response

Thank you for your letter and for the reviewers’ expert comments on our manuscript entitled “Construction and Swelling Properties of Thermosensitive N-isopropyl Acrylamide Microspheres with Controllable Size” (ID: materials-556092). All those comments are quite valuable and very helpful for further revising and improving the present manuscript and also guiding our future researching focuses. According to these expert comments, we have made the corresponding corrections carefully as follows. Certainly, these corrections do not influence the content and researching intention of the manuscript.

Thank you for your attention and kind help.

Best wishes

Liang Wang

Reviewer 3 Report

The work is interesting, can be published. We use a similar approach to obtain hollow microspheres with other objects. The diameter of the resulting microspheres is 150-200 nm.

Author Response

Thank you for your letter and for the reviewers’ expert comments on our manuscript entitled “Construction and Swelling Properties of Thermosensitive N-isopropyl Acrylamide Microspheres with Controllable Size” (ID: materials-556092). All those comments are quite valuable and very helpful for further revising and improving the present manuscript and also guiding our future researching focuses. According to these expert comments, we have made the corresponding corrections carefully as follows. Certainly, these corrections do not influence the content and researching intention of the manuscript.

Thank you for your attention and kind help.

Best wishes

Liang Wang

Response to Reviewer 3 Comments

Point 1: English language and style are fine/minor spell check required. The work is interesting, can be published. We use a similar approach to obtain hollow microspheres with other objects. The diameter of the resulting microspheres is 150-200 nm.

Response 1: According to the Reviewer’s comments, we have made the corresponding corrections carefully in the manuscript. Certainly, these corrections do not influence the content and researching intention of the manuscript.

Round  2

Reviewer 2 Report

Authors have responded all comments and suggestions. Indeed, they include in the revised version literature that support they results and conclusions. I consider that now the manuscript is suitable to be accepted in Materials in the present form

This manuscript is a resubmission of an earlier submission. The following is a list of the peer review reports and author responses from that submission.

Round  1

Reviewer 1 Report

This manuscript describes the synthesis of thermosensitive poly(N-isopropylacrylamide) (PNIPAM) with controllable size. The experiment is well designed and the results which are argued by author are interesting. However, there are several problems in this manuscript. In my opinion, this might be interesting after more improvement. Authors should clarify the following issues in order to strengthen this manuscript.

1. The authors provided the FTIR spectra of homogeneous microspheres and hollow microspheres in Figure 1. The FTIR spectrum of homogeneous microspheres is slightly different from that of hollow microspheres. More detailed explanations on the difference of the FTIR spectra are required for the better understanding. In addition, the authors need to deconvolute the FTIR spectra for more accurate analysis.

2. It is difficult to discern the uniformity of the particle shape and size by TEM images in Figure 1. Moreover, it seems that the TEM images are out of focus. High-quality TEM images should be provided. Furthermore, it seems there is something between the particles.

3. The authors controlled the particle size by adjusting the temperature from 60 to 90 °C. What if the temperature is lower than 60 °C or higher than 90 °C?

4. In Figure 2a, any tendency of the particle diameter depending on the temperature is not observed. More scientific and clear explanations are required.

5. The authors mentioned that the stirring speed had a great effect on the size uniformity of microspheres. To clarify this, the authors should provide data on the effect of stirring speed on the size uniformity.

6. In Figure 4, the swelling degree of the microspheres with controlling the reaction temperatures, initiator dosages, reaction time, and stirring speed are provided. For each case, the fixed conditions should be specified.

7. Lastly, as a minor point, there are some typing errors in the manuscript. For instance, there are some cases where hyphen is misused, as follows.

  “16.4~0.1” → “16.4–0.1”

  “1000 r min -1” → “1000 r min–1

Accordingly, I recommend this manuscript for publication in Materials after major revision.

Reviewer 2 Report

I can not recommend publishing the manuscript version for the following reasons:

- > Figure 1. Chemical formulas (a) and (b) are incorrect:

Methyl methacrylate (MAA) means methyl ester of methacrylic acid

In addition, the double bonds are completely absurd

-> The manuscript is like a laboratory journal. There are no explanations, for example why the amount of initiator influences the particle diameter